# Rainfall and Temperature Influences on Childhood Diarrhea and the Effect Modification Role of Water and Sanitation Conditions: A Systematic Review and Meta-Analysis

**DOI:** 10.3390/ijerph21070823

**Published:** 2024-06-24

**Authors:** Gorfu Geremew, Oliver Cumming, Alemayehu Haddis, Matthew C. Freeman, Argaw Ambelu

**Affiliations:** 1Department of Environmental Health Science and Technology, Jimma University, Jimma P.O. Box 378, Ethiopia; a_had12@yahoo.com; 2Department of Disease Control, Faculty of Infectious Tropical Disease, London School of Hygiene and Tropical Medicine, London WC1E 7HT, UK; oliver.cumming@lshtm.ac.uk; 3Gangarosa Department of Environmental Health, Rollins School of Public Health, Emory University, Atlanta, GA 30322, USA; matthew.freeman@emory.edu; 4Division of Water and Health, Ethiopian Institute of Water Resources, Addis Ababa University, Addis Ababa P.O. Box 1165, Ethiopia; aambelu@yahoo.com

**Keywords:** rainfall, temperature, under-five children, diarrhea, drinking water source, sanitation condition

## Abstract

The latest report from the Intergovernmental Panel on Climate Change (IPCC) highlighted the worsening impacts of climate change. Two climate factors—temperature and rainfall uncertainties—influence the risk of childhood diarrhea, which remains a significant cause of morbidity and mortality in low- and middle-income countries. They create a conducive environment for diarrhea-causing pathogens and overwhelm environmental prevention measures. This study aimed to produce comprehensive evidence on the association of temperature and rainfall variability with the risk of childhood diarrhea and the influence of water and sanitation conditions on those associations. We conducted a systematic review and meta-analysis using the Preferred Reporting Items for Systematic Review and Meta-Analysis (PRISMA) approach. Records published in English from 2006 to 2023 were searched on 8 January 2024 via PubMed, EMBASE, ScienceDirect, Scopus, the Cochrane Library, and Google/Google Scholar using comprehensive search terms. We assessed studies for any risk of bias using the Navigation Guide and rated the quality of the evidence using the GRADE approach. The heterogeneity among estimates was assessed using I-squared statistics (I^2^). The findings of the analysis were presented with forest plots using an incidence rate ratio (IRR). A meta-analysis was conducted on effect modifiers (water supply and sanitation conditions) using a random effects model with a 95% confidence interval (CI). The statistical analyses were conducted using R 4.3.2 software and Review Manager 5.3. A total of 2017 records were identified through searches, and only the 36 articles that met the inclusion criteria were included. The analysis suggests a small positive association between increased temperature and the occurrence of under-five diarrhea, with the pooled IRR = 1.04; 95% CI [1.03, 1.05], at I^2^ = 56% and *p*-value < 0.01, and increased rainfall and U5 diarrhea, with IRR = 1.14; 95% CI [1.03, 1.27], at I^2^ = 86% and *p*-value < 0.01. The meta-analysis indicated a positive association between unimproved latrine facilities and drinking water sources with a rainfall-modified effect on U5 diarrhea, with IRR = 1.21; 95% CI [0.95, 1.53], at I^2^ = 62% and *p*-value = 0.03. We found that an increase in mean temperature and rainfall was associated with an increased risk of childhood diarrhea. Where there were unimproved latrine facilities and drinking water sources, the increase in mean rainfall or temperature would increase the incidence of childhood diarrhea. The results of this review help in assessing the effectiveness of current intervention programs, making changes as needed, or creating new initiatives to lower the prevalence of childhood diarrhea.

## 1. Introduction

In a rapidly changing world, environmental variability due to climate change has been one of the major challenges to human health and growth, particularly children’s health [1,2]. It is estimated that from 2030 to 2050, climate change could cause approximately 250,000 additional deaths per year from malnutrition, malaria, diarrhea, and heat stress [3]. Worldwide, the burden of environmental disease is much greater for children than adults, particularly under-five-year-olds [2]. According to climate change evidence, among the disease burdens occurring in the world due to climate change, 88% fall on children under the age of five [4]. Moreover, the WHO evidence suggests that 48,000 deaths from diarrheal disease will occur annually by 2050 due to climate change [5].

Climate change affects children’s health through more frequent and intense heat waves, decreasing water quality and quantity, food shortages, and greater exposure to toxicants [6]. The disruption of the water sources and increasing temperatures due to global climate change are risk factors for increased water- and foodborne illnesses, leading to diarrhea and dehydration among children [7]. Temperature variability is one of the climate change effects that is estimated to influence diarrhea incidence by creating a conducive environment for pathogens to survive or multiply within the environment. Globally, temperatures are projected to rise by 2 °C and may increase by 4.8 °C in 2100 unless the current trends in emissions are changed [8].

Under-five children are especially susceptible to environmental factors, particularly contaminated food and drinking water, which are affected by the variability of temperature and rainfall events [9].

Globally, the burden of diarrheal disease has been one of the leading causes of morbidity and mortality among under-five children, even though different preventive measures have been implemented. In 2019, diarrheal illness claimed the lives of 370,000 children under the age of five worldwide [10]. This is also supported by the evidence summarized from 15 high-burden focus countries: every day, about 1200 young children die due to diarrhea [11].

To identify the reasons for the increasing burden of diarrheal disease, in the recent decade, studies have been conducted on the association between climate factors and diarrheal disease. Among these parameters, temperature and rainfall are the most important ones on which diarrhea-causing pathogens are mainly dependent [12]. There were systematic reviews and meta-analyses that were conducted on the impact of rainfall or temperature on diarrhea in people of all ages [13,14,15]. Their conclusions show that changes in temperature due to global climate change may affect or are already affecting diarrheal disease incidence, and bacterial and parasitic diarrhea were more common during rainy seasons. However, they did not assess the pooled effect estimate on the impact of temperature or rainfall on U5 diarrhea, even though they included articles conducted on the influence of either rainfall or temperature on U5 diarrhea.

As compared to adults, children, particularly young ones, are uniquely vulnerable to climate change, in part because of the natural physiology of their developing and growing bodies [16]. They have less control over their physical environments, less knowledge about the health effects of climate change, and less ability to remove themselves from harm [17]. Thus, conducting a systematic review is crucial to assessing the extent of the burden that climatic variability places on under-five children.

There is no existing systematic review of the evidence on the association between rainfall or temperature and under-five diarrheal disease. Rainfall and temperature variability by themselves do not directly cause diarrheal disease. This is because there should be drinking water and sanitation conditions that are favorable for the pathogens to spread in conditions related to rainfall and temperature.

Thus, the review objective was to answer the following research questions: What relationship exists between changing rainfall patterns and the occurrence of childhood diarrhea? What relationship exists between changes in temperature and the occurrence of childhood diarrhea? Where there are poor water and sanitation conditions, what is the relationship between rainfall and childhood diarrhea, as well as the relationship between temperature and childhood diarrhea?

Therefore, based on the above research questions, this systematic review aimed to assess and produce comprehensive evidence on the association of rainfall and temperature with childhood diarrhea, as well as the influence of water and sanitation conditions on the relationship of temperature and rainfall with childhood diarrhea.

## 2. Methods

This systematic review and meta-analysis were conducted using the Preferred Reporting Items for Systematic Review and Meta-Analysis (PRISMA) approach. The review article was registered on the PROSPERO checking system with receipt number CRD42023447638.

### 2.1. Eligibility Criteria

Articles that met the following criteria were included in the review: Articles that assessed the impact of rainfall and/or temperature on under-five diarrhea were included. Articles that assessed the impact of rainfall and/or temperature on diarrhea in people of all ages but produced separate analyses (effect estimates) on U5 children were included. In addition, articles conducted on the influence of water and sanitation conditions on the relationship between rainfall and/or temperature on U5 diarrhea were included. Articles of any research design (either observational or experimental) with a minimum six-month study period were included. In order to have sufficient evidence for our review, we included records that were published in English from January 2006 to December 2023, with no limit to the study country. Because the review was quantitative, published articles with effect estimates were included.

In order to keep the precision of the effect estimate, we did not convert other scales of measurements such as the correlation coefficient (r), standard mean difference (SD), and the like into the common incidence rate ratio (IRR). In addition, studies whose effect estimates could not be converted to the incidence rate ratio (IRR) were excluded from the analysis.

### 2.2. Information Sources

The review title was structured according to the PECO (Population, Exposure, Comparator, Outcome) statement. This is because a clearly framed question creates the structure and delineates the approach to defining research objectives and conducting systematic reviews [18]. In addition, it informs the study design or inclusion and exclusion criteria for a review, as well as facilitates the interpretation of the directness of the findings based on how well the actual research findings represent the original question [19]. Thus, our review question contained population (under-five children); exposures (temperature and rainfall variability); comparators (water and sanitation conditions: type, availability, accessibility, and quality); and an outcome variable (diarrhea). We searched databases such as PubMed, EMBASE, ScienceDirect, Scopus, Cochrane Library, Google, and Google Scholar for the relevant records using comprehensive search terms.

### 2.3. Search Strategy

The search strategy was developed in line with the research question formulated according to the PECO statement. We conducted a pilot test for the sensitivity and/or comprehensiveness of the search strategy against benchmark papers. In addition, to have comprehensive search terms that provide all relevant records, we searched for records by exchanging search terms combined with the Boolean operators. Finally, we used the main search terms combined as follows: “Rain* OR temperature AND “water quality” OR sanitation AND diarrhea OR diarrhoea AND child*”. The final search for records through the databases was conducted on 8 January 2024. The search strategies and combined research terms used for databases are described in an Excel spreadsheet; see the Appendix A. In addition, we assessed the references listed under relevant articles for inclusion in the review.

### 2.4. Study Selection

First, two reviewers (GG and AA) searched, identified, and screened articles by reading titles and abstracts. For each article included in the systematic review and meta-analysis, a consensus was reached among the reviewers (GG, AA, and AH). Prior to the data extraction, the reviewers GG and AA adopted the standardized Excel sheet, and consensus was reached with the support of the third reviewer, AH.

### 2.5. Data Extraction

The authors (GG, AA, and AH) extracted data such as first author, publication year, study country, study design, study duration, number of U5 diarrhea cases, the definition of exposures and outcome variables, cut-off points for rainfall and temperature, source of drinking water, WASH factors, effect estimates, the scale of effect measures, adjusted covariates, lag type, statistically analyzed model, major findings, study country by region, and income level from the articles.

### 2.6. Data Items

#### 2.6.1. Outcome Variable

Diarrhea was used as an outcome variable: Articles that reported diarrheal disease among U5 children in different ways, such as diarrhea, diarrhoea, acute watery diarrhea, gastrointestinal infection, bacterial dysentery, rotavirus infection, etc., that operationalized it according to the WHO definition of diarrheal disease, ‘three or more loose or liquid stools per day’ [20], were included in the review. We included studies that measured childhood diarrhea with different recall periods but in accordance with the WHO definition.

#### 2.6.2. Exposure Variables

In this review, temperature (an exposure variable) was defined as the mean, so that the temperature reported in different ways in the selected articles was approximated to the mean value. Therefore, the term ‘mean temperature’ was used to determine the association between childhood diarrhea and temperature.

Rainfall, another exposure variable, was defined as the monthly mean so that the rainfall reported in different ways in the selected article was approximated to the monthly mean value. Therefore, the term ‘mean rainfall’ was used to determine the association between childhood diarrhea and rainfall.

The ‘water conditions’ that were considered in this review refer to the type, accessibility, availability, and quality of drinking water sources that households use.

The phrase ‘sanitation conditions’ that were considered in this review included the availability, type, and accessibility of latrines as well as household and compound sanitation.

### 2.7. Risk of Bias Assessment

Two reviewers, GG and AA, independently assessed the risk of bias for each study using the Navigation Guide tool for systematic review methodology [21]. We preferred to use this risk of bias assessment tool because there was no new or updated tool designed for time-series count data. This tool consists of eight risk domains: sample representativeness, blinding, exposure assessment, confounding, incomplete data, selective outcome reporting, conflict of interest, and other sources of bias. Using pre-determined criteria, the reviewers rated the risk of bias domain as “not applicable”, “low risk of bias”, “probably low risk of bias”, “high risk of bias,” or “probably high risk of bias,” so that the final result was decided through the agreement of three reviewers (GG, AA, and AH); see Appendix A. Furthermore, we also showed this finding by categorizing it as unclear, low risk of bias, or high risk of bias using the Cochrane risk of bias graph and summary, as shown in Figure 1).

The authors’ view judgements about each risk of bias item for each included study are presented as a risk of bias summary (see Appendix A).

### 2.8. Rating the Quality of Evidence

The same reviewers assessed the overall quality of evidence for outcomes across the included studies using the GRADE framework, adapted for environmental and occupational health [22]. Hence, all studies included in the review were observational; we first rated the evidence quality as low and then considered eight critical components to upgrade or downgrade the evidence quality. In the end, we rated the body of evidence for primary outcomes using four levels of quality: “very low”, “low”, “moderate”, or “high” quality. The overall process of certainty in evidence assessment is presented in the evidence summary table; see the Appendix A.

## 3. Data Analysis

Prior to the analysis, the effect estimate reported in another ratio scale measure in each selected study was converted into a common ratio scale measure, the incidence rate ratio measure (IRR). We followed different approaches to convert the effect estimate presented in various ratio scale measures into the incidence rate ratio. In a given study, if the odds ratio (OR) of people exposed to a particular exposure was less than 10%, it was roughly equivalent to the IRR. In a study in which the relative risk (RR) was determined from a rare exposure or from evenly spaced time-series data, the value of the RR is approximately equivalent to the IRR. Similar assumptions were held for other effect estimates, such as the prevalence ratio (PR) and hazard ratio (HR).

Moreover, for articles that reported the influence of temperature or rainfall on childhood diarrhea in lag categories, we averaged the effect estimate and used it in accordance with the predictor variables.

Because the data extracted from the articles were effect estimates with confidence intervals (CIs), we determined the standard error (SE) from the CI using the RevMan software. Then, the natural logarithm of the incidence rate ratio (LogIRR) and corresponding SE were determined using the generic inverse variance data (effect estimate and standard error). Finally, the inverse variance statistical method was used to produce the analyses with LogIRR, SE, and weight (the contribution of each effect estimate to the pooled effect size). Thus, the pooled effect estimates of the exposure–outcome associations were reported via a random effect model with the IRR. For some analysis, to increase the visibility of forest and funnel plots, we used R 4.3.2 software with the meta package consisting of metagen, forest, and funnel functions.

In this review, we conducted separate analyses on the influence of rainfall and temperature on childhood diarrhea. After excluding the studies that did not consider the lag effect, we assessed the presence of variation in the pooled effect estimate produced on the influence of temperature on childhood diarrhea. Similarly, we looked at the pooled effect estimate of temperature after excluding studies with a cross-sectional study design, then studies with a cross-sectional design, and those that did not consider the lag effect. The reason for this assessment was to discern whether there is significant variation due to the lag effect and study design or not.

In this review, in addition to the systematic reviews, we conducted subgroup, sensitivity, and meta-analyses using RevMan software. A subgroup analysis was conducted to determine the status of temperature’s influence on childhood diarrhea by country income level.

The sensitivity analyses were conducted on the influence of rainfall and temperature, separately, on childhood diarrhea. The reason is that both exposures are sensitive to each other, so the findings of rainfall on childhood diarrhea can be influenced by the temperature and vice versa. In order to control these confounders’ effects on the outcome findings, some studies tried to adjust the model during the analysis, even though it is not possible to control them completely. It is assumed that the pooled effect estimate produced from articles that assessed both exposures is more reliable.

We assessed studies with effect estimates that could contribute high heterogeneity to the pooled effect size using the one-living-out method during each analysis. The heterogeneity contributed via each article was reported using the I-squared (I^2^) statistic, and the publication bias was shown via funnel plots.

In the end, we assessed the influence of water and sanitation conditions on the relationship between temperature and rainfall on childhood diarrhea in meta-analyses using a random effect model. We conducted separate stratified meta-analyses because the extent of the influence of water and sanitation conditions might vary based on temperature and rainfall associations with diarrhea. In addition, the effect modifiers, in this case the water and sanitation variables, were different and their influence might also be varied.

## 4. Results

A total of 2017 records were identified via searches in selected major e-databases and Google/Google Scholar, of which only 430 papers were selected after being reviewed through titles and abstracts. A total of 231 articles were identified for full-text review; 36 articles with effect estimates were selected for systematic review and meta-analysis (Table 1). For the meta-analysis that was conducted on the influence of water and sanitation conditions, eight articles with eleven effect estimates were retrieved, as shown in the PRISM diagram; Figure 2. Furthermore, these studies (36) with selected main characters are presented in Appendix A.

Of these, 12 examined only the impact of temperature (in Appendix A), 8 examined the impact of rainfall (in Appendix A), and 16 looked at the influence of both temperature and rainfall on childhood diarrhea (in Appendix A).

In this review, two exposures of interest—rainfall and temperature—were separately analyzed. In the analysis of the association between temperature and childhood diarrhea, a total of 29 articles were selected (see Appendix A). When the effect estimates of these articles were analyzed, the pooled effect estimates of temperature on childhood diarrhea became IRR = 1.05, 95% CI [1.02, 1.09] (see Appendix A; forest plot Appendix A and funnel plot Appendix A). This indicates that there was a 1.05-fold higher incidence of diarrhea among U5 children for a 1 °C increase in mean temperature. We re-analyzed the model by removing effect estimates, one at a time, and identified studies that contributed to high heterogeneity. After removing these studies, the overall effect estimate of temperature on childhood diarrhea became IRR = 1.04, 95% CI [1.03, 1.05], with I^2^ = 56% (*p* < 0.01), as shown in Figure 3.

Furthermore, we analyzed the model by removing effect estimates from cross-sectional studies and those that did not consider the lag effect; however, there was no change in the overall effect estimate; IRR = 1.05, 95% CI [1.03, 1.08], with I^2^ = 55%. Under the temperature–diarrhea assessment, a subgroup analysis based on the income level of countries was conducted. The finding shows that the pooled effect estimates on the association between temperature and childhood diarrhea varied across country income categories, from a low association in high-income countries to a high association in low-income countries (see the Appendix A). We also conducted a sensitivity analysis for 16 articles that assessed both temperature and rainfall influence on U5 diarrhea. As a result, the overall effect estimates produced on the association between temperature and childhood diarrhea was IRR = 1.05, 95% CI [1.01, 1.10], with I^2^ =100% (*p* < 0.0001); see the Appendix A provided in Appendix A.

We determined the association between rainfall and childhood diarrhea using the same approach applied to temperature. A total of 21 articles that were assessed for the influence of rainfall on childhood diarrhea were identified (see Appendix A). In the first analysis, the pooled effect estimates of 21 articles became IRR = 1.01 [1.00, 1.01]. This shows that there was a 1.01-fold higher incidence of diarrhea among U5 for a 1mm increase in mean rainfall. Finally, after removing those with high heterogeneity, a total of 16 articles were included, so that the pooled effect estimates of rainfall on diarrhea became IRR = 1.14 [1.03, 1.27], with I^2^ = 87% (*p* < 0.01) as shown in Figure 4.

Of the total studies that examined the relationship between rainfall and diarrhea among U5 children, 14 evaluated their relationship in the rainy season. Most of them reported that childhood diarrhea cases increased during the rainy season or during periods of heavy rain following dry days.

Prior to conducting the meta-analysis, we extracted effect estimates of rainfall and temperature from included articles that were produced after interacting with the effect modifiers during analysis (see Appendix A). The effect modifiers we used in these meta-analyses were high access to piped (improved) drinking water sources, access to the public sewerage network, unimproved latrine facilities, and low access to piped drinking water sources.

Since the effect modifiers were different, their influence on the association between temperature and childhood diarrhea was presented separately in a stratified meta-analysis. Accordingly, two effect modifiers categorized as improved (high access to piped drinking water sources and high access to the public sewerage network) and two effect modifiers categorized as unimproved (unimproved latrine facilities and low access to piped drinking water sources) were reported in the forest plot, as shown in Figure 5.

Even though not pooled, the forest plot shows that unimproved effect modifiers had a positive influence on the association between temperature and childhood diarrhea (see the Appendix A for the funnel plot: Appendix A).

Similarly, another meta-analysis conducted on six studies indicated that the overall effect estimate of the association between rainfall and childhood diarrhea among study populations with poor access to water and sanitation was IRR = 1.01, 95% CI [1.00, 1.01], with I^2^ = 73% (*p* = 0001) (see the Appendix A). After excluding one study with high heterogeneity, the pooled effect estimates became = 1.21, 95% [0.95, 1.53], I^2^ = 62% (*p* = 0.03), as shown in Figure 6.

Heterogeneity: The potential heterogeneity that could be introduced via each estimate on the overall effect estimate was assessed using I^2^ and the *p*-value in the forest plot. Accordingly, most analyses show that there was considerable heterogeneity, even though its level varied from low to high in percent. In addition, the heterogeneity contributed via each effect estimate (outliers) was checked by removing them one by one from the model. As a result, those articles that could introduce considerable heterogeneity were shown in the analysis. Based on this, we conducted sensitivity and subgroup analyses.

Publication Bias: In this review, publication bias was assessed by looking at the distribution of effect estimates on the funnel plot. The plot shows the SE (log [incidence rate ratio]) on the y-axis against the incidence rate ratio (IRR) on the x-axis. Accordingly, the findings show the effect estimates were asymmetrically distributed over the plot. However, depending on the analysis, the distributions ranged from slightly to considerably asymmetric. Funnel plots of each pooled effect estimate produced for pre-defined exposure–outcome associations are provided in the Appendix A.

## 5. Discussion

In this systematic review, we assessed the influence of temperature and rainfall on under-five diarrheal disease as well as the effect size modification potential of water supply and sanitation conditions. The analyses produced pooled effect estimates that revealed there is a positive association between rainfall/temperature and childhood diarrhea.

This systematic review suggests that an increase in rainfall and temperature would increase the incidence of childhood diarrhea. This is consistent with the previous review meta-analysis finding that an increase in temperature and rainfall predicts an increase in cryptosporidiosis, a primarily waterborne diarrheal illness [59]. Our findings strengthen the evidence of an overview of the systematic review [60] and clustered random trial [61] that reported temperature and rainfall as the main contributors to the occurrence of diarrheal disease. In addition, the review of studies conducted on the association between global climate change and children’s health supports our findings by summarizing that the change in temperature and rainfall increased diarrheal diseases [62]. Our findings also support the implications that climate change will likely increase children’s diarrhea through more heavy rainfall and rising water temperatures, particularly in the developing world [8,63]. The reviews of scientific evidence from Asian cities [64] and Ethiopia [65] are in line with our findings. However, systematic reviews and meta-analyses that were conducted on the association of temperature and rainfall on rotavirus diarrhea reported, in contrast, that diarrhea increases in colder or drier seasons rather than in higher mean temperatures or rainfall conditions [66].

In this systematic review, the pooled effect estimates of 29 studies indicate a significant positive association between temperature and childhood diarrhea. This could be because the diarrhea-causing pathogen could multiply with increasing temperatures. This is consistent with the previous systematic review’s finding that there is an association between ambient temperature and diarrhea, even though it was not based on diarrhea in under-five children [13,67]. Another systematic review analysis [6] and review of studies [68] on the impact of ambient temperature on children’s health support our findings, revealing that temperature is a significant contributor to the occurrence of gastrointestinal diseases among children. However, evidence from a systematic review reflected that, except for viral diarrhea cases, there is a positive association between ambient temperature and diarrheal diseases [15].

In this systematic review, a subgroup analysis that was conducted on the association between temperature and childhood diarrhea based on the income level of countries showed positive associations among all categories, with the highest being in low-income countries. This is consistent with the previous systematic review that showed a positive association between temperature and diarrhea in all income categories of countries [13].

In this systematic review, the pooled effect estimates of 16 studies indicate a significant positive association between rainfall and childhood diarrhea. In the evidence associated with rainfall, there are two different concepts. The first suggests that heavy rain decreases diarrhea by cleaning pathogens from the environment; it is low rainfall that exaggerates the pathogens [69]. The second piece of evidence says that heavy rainfall increases the incidence of diarrhea by flushing pathogens into drinking water sources, especially in areas where access to piped water and improved latrines is limited. Our findings support the idea that an increase in rainfall would increase the incidence of childhood diarrhea. This is also supported by the previous systematic review that suggests diarrhea could occur with increased rainfall [14]. In addition, the present study is supported by evidence that summarized anomalously wet conditions increase the risk of diarrhea in humid, subtropical regions [70]. However, our finding contrasts with the evidence found in the global cross-sectional study of diarrhea incidence in children under 5; there is a negative association between increased rainfall and the occurrence of diarrhea [71]. It is here that assessing the pooled modification effect of WASH factors on the relationship between rainfall, temperature, and childhood diarrhea is needed.

Our meta-analysis suggests that where there were no or low-improved WASH factors, the increased mean temperature or rainfall would increase the incidence of childhood diarrhea. This meta-analysis produced the pooled effect estimates in terms of WASH variables: latrine facilities and drinking water sources. This is consistent with the finding of the study that shows that when mean temperature and rainfall increase, households using unimproved sanitation or unimproved water have more cases of diarrhea [43]. On the other hand, the interventional study reported that when there were co-occurrence shifts in temperature and precipitation, the prevalence of childhood diarrhea increased by half in areas with unimproved WASH [72].

The pooled effect estimate of the interaction of unimproved latrine and rainfall on childhood diarrhea was IRR = 1.29 [0.87, 1.9]. This is supported by the evidence that shows that following heavy rain, there is an increased risk of gastrointestinal disease, particularly in areas with poor hygiene [15,73]. Furthermore, another piece of evidence reflected that the change in rainfall posed a risk to unimproved sanitation services, which in turn led to infectious diseases, including diarrhea [74]. The present evidence demonstrates that even though the pooled effect estimate is based on a few studies [28,49], in areas with poor or no toilet usage, heavy or low rainfall could increase the incidence of childhood diarrhea.

The other finding of this review shows that the pooled effect estimate of the interaction of unimproved drinking water sources and rainfall [28,30,42] on childhood diarrhea was IRR = 1.22 [0.89, 1.68]. This is consistent with the finding that shows heavy precipitation could adversely contaminate drinking water supplies, which in turn results in childhood diarrhea [75]. This could be heavy rain that stirs up sediment of diarrhea-causing pathogens in water, leading to the accumulation of large pathogens [76].

In addition, our meta-analysis suggests that where there is low access to improved toilets and drinking water sources, there is a positive pooled effect estimate between temperature and childhood diarrhea, with IRR = 1.06 [1.00, 1.13]. This is supported by other research that indicated high access to piped water and the public sewerage network could decrease the incidence of diarrhea with an increased mean temperature [47].

Furthermore, our meta-analysis findings are consistent with the previous meta-analysis of WASH interventions that revealed basic sanitation services with sewer connections resulted in a reduction in diarrhea compared to unimproved sanitation [77]. However, this evidence was not based on articles on under-five diarrhea and did not take into account temperature and rainfall variables. Similarly, other systematic reviews conducted on the impact of basic sanitation on diarrhea support our meta-analysis findings [78]. In general, our meta-analysis of a few studies indicates that unimproved latrine and drinking water sources exaggerate the incidence of childhood diarrhea in cases of increased temperature and rainfall.

The review study has many strengths: Initially, it has tried to address the recommendations put forth by the prior systematic review studies: the age-specific review and the potential of WASH factors on the effect modification [13,14,79]. Accordingly, in this review, articles were searched systematically and analyzed to determine the overall effect of temperature and rainfall on childhood diarrhea. The WASH factors operationalized in this review were assessed for their potential to modify the effects of temperature and rainfall on childhood diarrhea.

There were also limitations to the review: As has been the case in many review studies, limiting the review to English-language published articles is a weakness of this review. The evidence shows that rainfall and temperature variability have become major sources of infectious diseases, including diarrhea, in developing countries. However, few studies were included from low-income countries. This indicates that there is a considerable publication bias. Confounding, both manageable and unmanageable, could have an impact on the effect estimates of the identified articles. Even though studies evaluating the seasonal variation in diarrhea attributed to rainfall were identified, one of the review’s weaknesses is that it did not generate a pooled effect estimate based on seasonal variation.

## 6. Conclusions

We found that an increase in mean temperature and rainfall was associated with an increased risk of childhood diarrhea. In addition, the status of the WASH condition plays a major role in influencing the relationship between temperature and rainfall in childhood diarrhea. Where there were unimproved latrine facilities and drinking water sources, the increase in mean rainfall or temperature would increase the incidence of childhood diarrhea. To our knowledge, this systematic review and meta-analysis is the first to produce comprehensive evidence on the impact of rainfall and temperature on childhood diarrhea as well as the influence of unimproved drinking water sources and sanitary facilities.

## Figures and Tables

**Figure 1 ijerph-21-00823-f001:**
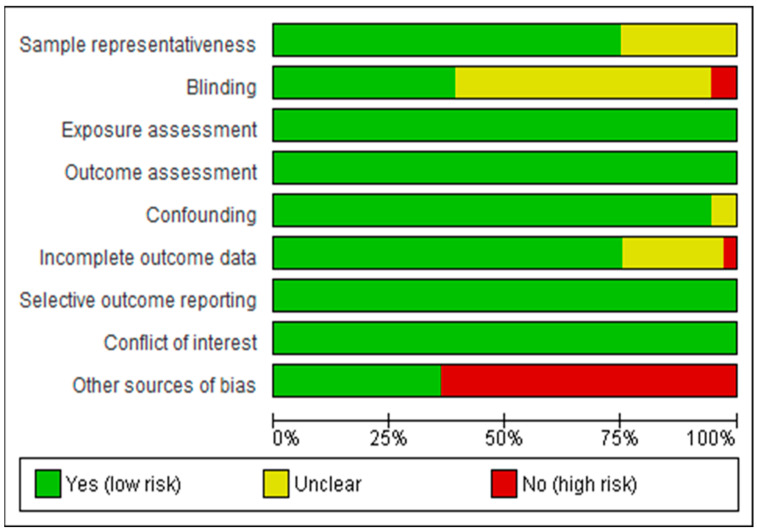
Risk of bias graph: review authors’ judgements about each risk of bias item presented as percentages across all included studies.

**Figure 2 ijerph-21-00823-f002:**
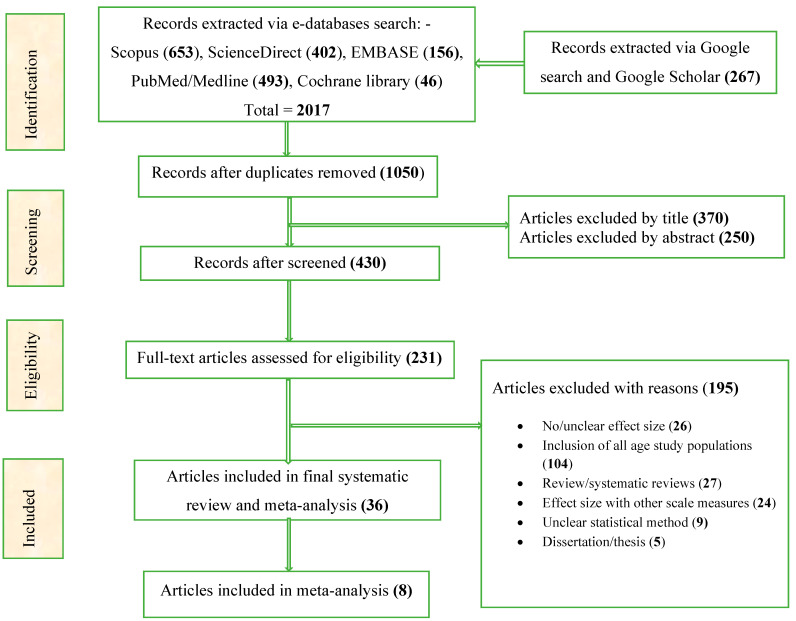
The PRISMA diagram of the study selection for a systematic review and meta-analysis on the association between rainfall and temperature on childhood diarrhea and their influence on water and sanitation factors.

**Figure 3 ijerph-21-00823-f003:**
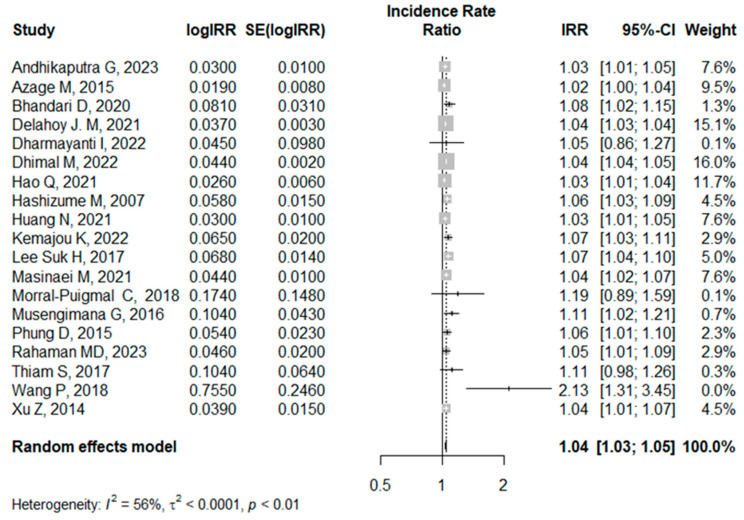
Forest plot of the systematic review of the association between temperature and childhood diarrhea after removing the articles that contributed to high heterogeneity (the funnel plot displaying the corresponding publication bias is shown in the Appendix A) [24,26,27,32,33,35,37,39,41,43,45,47,50,51,52,53,54,55,58].

**Figure 4 ijerph-21-00823-f004:**
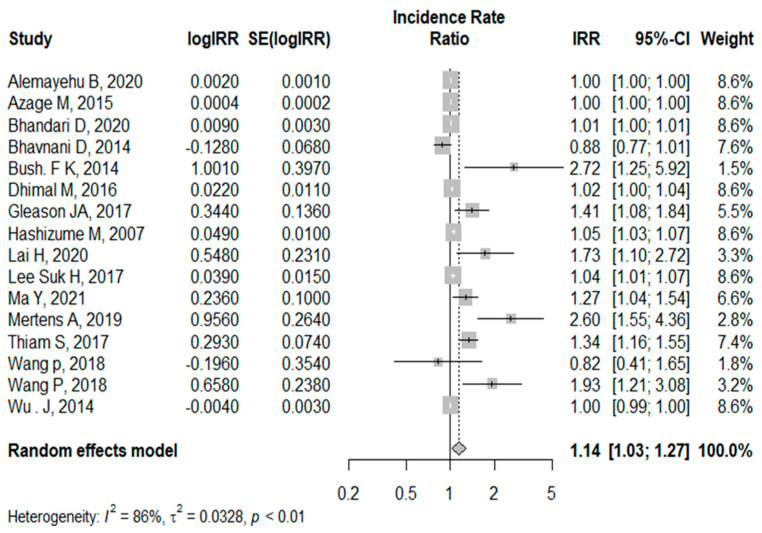
Forest plot of the systematic review of the association between rainfall and childhood diarrhea (the funnel plot of this analysis is shown in the Appendix A) [23,26,27,28,29,34,36,39,44,45,46,49,54,55,58].

**Figure 5 ijerph-21-00823-f005:**
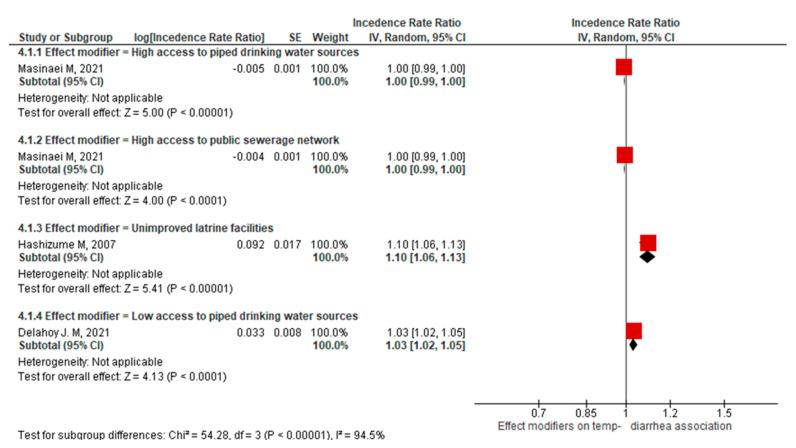
Forest plot of the meta-analysis on the modified association between temperature and childhood diarrhea due to water and sanitation conditions. Effect modifiers on temp-diarrhea association refers Effect modifiers on the temperature-diarrhea association [32,39,47].

**Figure 6 ijerph-21-00823-f006:**
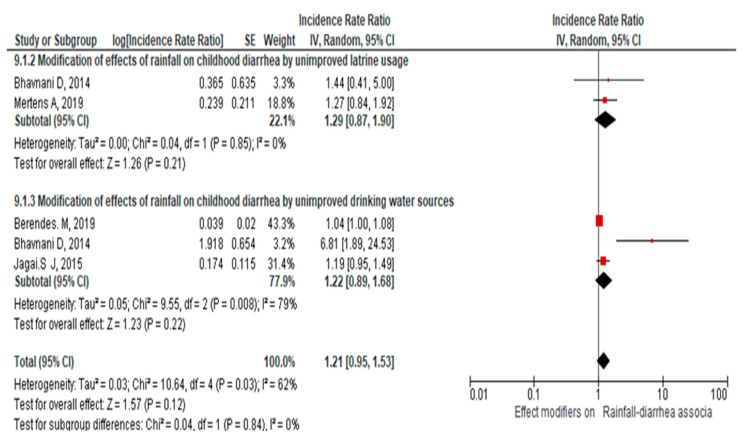
Forest plot of the meta-analysis on the modified effect of the association of rainfall on childhood diarrhea by water and sanitation conditions (see the Appendix A for funnel plot, Appendix A). Effect modifiers on Rainfall-diarrhea associa refers Effect modifiers on the Rainfall-diarrhea association [28,30,42,49].

**Table 1 ijerph-21-00823-t001:** Selected articles with main study characteristics for systematic review and meta-analysis on the effect of rainfall and temperature on childhood diarrhea and the modification effect of water and sanitation conditions: January 2006–December 2023 [23,24,25,26,27,28,29,30,31,32,33,34,35,36,37,38,39,40,41,42,43,44,45,46,47,48,49,50,51,52,53,54,55,56,57,58].

S. N	First Author, Year	Study Country	Study Design	Study Population	Outcome	Exposure Variable/s	Lag Unit	Association	Income
01	(Alemayehu et al., 2020) [23]	Bench Maji, Ethiopia	Retrospective study	<5 age children	Diarrhea	Rainfall, Temperature	None	Positive	LI
02	(Andhikaputra et al., 2023) [24]	Taiwan	HMIS data	<5 age children	Diarrhea	Temperature	Month	Positive	UM
03	(Atchison et al., 2010) [25]	Great Britain & Netherlands	Surveillance study	<5 age children	Rotavirus diarrhea	Rainfall, Temperature	week	Positive	HI
04	(Azage et al., 2015) [26]	Amhara, Ethiopia	Retrospective study	<5 age children	Diarrhea	Rainfall, Temperature	Month	Positive	LI
05	(Bhandari et al., 2020) [27]	Kathmandu, Nepal	Retrospective study	<5 age children	Diarrhea	Rainfall, Temperature	Month	Positive	LM
06	(Bhavnani et al., 2014) [28]	Borbo area, Ecuador	Serial case–control	<5 age children	Diarrhea	Rainfall	Day	Negative	UM
07	(Bush et al., 2014) [29]	Chennai city, India	Time-series study	<5 age children	Diarrhea	Rainfall	Day	Positive	LM
08	(Berendes et al., 2019) [30]	Vellore, India	Cohort study	<5 age children	Infectious Diarrhea	Rainfall	None	Positive	LM
09	(D’Souza, 2008) [31]	Brisbane, Australia	Time-series study	<5 age children	Rotavirus diarrhea	Temperature	Week	Positive	HI
10	(Delahoy et al., 2021) [32]	Peru	Ecological study	<5 age children	Diarrhea	Temperature	Week	Positive	UM
11	(Dharmayanti et al., 2023) [33]	Kalimantan, Indonesia	Ecological time-series study	<5 age children	Diarrhea	Rainfall, Temperature	Month	Positive	UM
12	(Dhimal et al., 2016) [34]	Kathmandu, Nepal	Retrospective study	<5 age children	Diarrhea	Rainfall, Temperature	None	Positive	LM
13	(Dhimal et al., 2022) [35]	Nepal	Ecological study	<5 age children	Diarrhea	Rainfall, Temperature	None	Positive	UM
14	(Gleason, 2017) [36]	New Jersey, USA	Bi-directional case-crossover	<5 age children	GI infection	Rainfall	Day	Positive	HI
15	(Q. Hao & Wang, 2021) [37]	Guangdong, China	Surveillance study	<5 age children	Infectious diarrhea	Temperature	Day	Positive	UM
16	(Y. Hao et al., 2019) [38]	Anhui, China	Ecological study	<5 age children	Bacillary dysentery	Temperature	Week	Positive	UM
17	(Hashizume et al., 2007) [39]	Dhaka, Bangladesh	Surveillance study	<5 age children	None-cholera diarrhea	Rainfall, Temperature	Week	Positive	LM
18	(Hashizume et al., 2008) [40]	Dhaka, Bangladesh	Surveillance study	<5 age children	Rotavirus Diarrhea	Temperature	Week	Positive	LM
19	(Huang et al., 2021) [41]	Jiangsu, China	Time-series study	<5 age children	Infectious diarrhea	Temperature	Day	Positive	UM
20	(Jagai, 2015) [42]	Massachusetts, USA	Ecological time-series study	<5 age children	GI illness diarrhea	Rainfall	Day	Positive	HI
21	(Kemajou, 2022) [43]	SSA	Demographic and health surveys	<5 age children	Diarrhea	Rainfall, Temperature	Month	Positive	LI
22	(Lai et al., 2020) [44]	New Zealand	Nested case-crossover	<5 age children	Diarrhea	Rainfall	Day	Positive	HI
23	(Lee et al., 2017) [45]	Kon Tum, Vietnam	Retrospective study	<5 age children	Bacillary dysentery	Rainfall, Temperature	Month	Positive	LM
24	(Ma et al., 2021) [46]	Chongqing, China	Time-series study	<5 age children	Bacterial dysentery	Rainfall	Day	Positive	UM
25	(Masinaei, 2022) [47]	Iran	Retrospective time-series study	<5 age children	Watery diarrhea	Temperature	None	Positive	LM
26	(Mengistie et al., 2022) [48]	Kersa, Ethiopia	A community-based longitudinal survey	<5 age children	Diarrhea	Rainfall	None	Positive	LI
27	(Mertens et al., 2019) [49]	Tamil Nadu, India	Prospective Cohort	<5 age children	Diarrhea	Rainfall, Temperature	Week	Positive	LM
28	(Morral-puigmal et al., 2018) [50]	Spain	Ecological time-series study	<5 age children	GI disease	Temperature	Day	Positive	HI
29	(Musengimana et al., 2016) [51]	Cape Town, South Africa	Cross-sectional study	<5 age children	Diarrhea	Temperature	Week	Positive	UM
30	(Phung et al., 2015) [52]	Can Tho City, Vietnam	Time-series study	<5 age children	GI infection	Rainfall, Temperature	Day	Positive	LM
31	(Rahaman et al., 2023) [53]	Bangladesh	Surveillance study	<5 age children	Diarrhea	Temperature	Day	Positive	LM
32	(Thiam et al., 2017) [54]	Mbour, Senegal	Retrospective study	<5 age children	Diarrhea	Rainfall, Temperature	Month	Negative	LI
33	(Wang et al., 2018) [55]	Hong Kong, China	Time-series study	<5 age children	Roto virus diarrhea	Rainfall, Temperature	Day	Negative	UM
34	(Wu et al., 2014) [56]	Matlab, Bangladesh	Cohort study	<5 age children	Diarrhea	Rainfall, Temperature	Day	Positive	LM
35	(Xu et al., 2013) [57]	Brisbane, Queensland	Time-series observational study	<5 age children	Diarrhea	Temperature	Day	Positive	HI
36	(Xu et al., 2014) [58]	Brisbane, Australia	Ecological study	<5 age children	Diarrhea	Temperature	Day	Positive	HI

HI—High Income, LI—Low Income, LM—Lower-Middle Income, UM—Upper-Middle Income.

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
