# Peer review of "Rainfall and Temperature Influences on Childhood Diarrhea and the Effect Modification Role of Water and Sanitation Conditions: A Systematic Review and Meta-Analysis"

_ijerph, 2024, doi:10.3390/ijerph21070823_

Round 1
Reviewer 1 Report
Comments and Suggestions for Authors
Please see attached Word document.

The writing is generally clear. A review of grammar and phrasing is needed, mostly for small changes such as changing "the" to "a", etc.
Reorganization of some paragraphs would be helpful. Currently some ideas are spread out in non-contiguous paragraphs.
Author Response
Dear, Reviewer,
please see the attachment
If you face line/line inconsistency while reviewing the revised manuscript, please find the highlighted test around.

Reviewer 2 Report
Comments and Suggestions for Authors
Dear Respectable Authors
Thank you for considering a significant area of research related to childhood diarrhea. You investigated the influences of rainfall and temperature, and also the effect modification role of water and sanitation conditions through a systematic review and meta-analysis. Your results are of interest but your manuscript needs some revisions as follows. I think these modifications will improve the quality of your manuscript and the way you report your results.
- Abstract, please remove all subheadings from the abstract considering the journal guidelines.
- Line 19, please remove the methods from the aim of the study and mention it on the methods. You can rewrite this line as follows: "This study aimed at ...
- Abstract, please insert the exact date of search.
- Abstract, please add more details regarding the way you conducted the meta-analysis including which model, CI, etc.
- Line 30, please replace "selected" with "included".
- Abstract, lines 30-34, please add p-value and I2 statistics for each estimate.
- Abstract, please remove lines 34-37. This issue is not as important to mention in the abstract. Please add this statement to the main text. You need to add a conclusion and a direct response to the aims of the study without statistical difficulties so that the general reader can understand it.
- In my opinion, the style you used in the text for citation is not based on the journal format. However, it is better to check the guidelines and enter the desired style of the journal if needed.
- Lines 52-3, I think you need to add a reference for this statement. Also, lines 69-70. The items inside the parentheses may be the same references, but they are displayed in this way due to the selection of an inappropriate style. There are similar incidents in other places of the text that need to be investigated.
- Lines 87-90, these statements are related to the methods section. Please remove it from here and add it to the methods.
- Line 103, 2.1. please separate this information considering the PRISMA. These are different parts of a systematic review. First, mention the information related to registration and guidelines. Then, add a subheading for "eligibility criteria" and a subheading for "information sources and search strategy". after that "study selection" with details.
- The Excel files included in the text are not available in the supplementary file. Check whether it was sent to the joyrnal or it is a mistake of the journal that did not put them on the site available for reiewers.
- Please merge2.2. and 2.3 under "eligibility criteria" (item 5 of the PRISMA 2020).
- Please add the exact date of search and search period and also the limit you used for search, What is you reason for limiting search from 2006?
- Please also add the full search strategy for all databases including the number of records.
- You have made a common mistake that unfortunately many writers make. The opposite of the inclusion criteria is not the exclusion criteria. This means that when you import English language articles, there are no other articles except English that we want to remove. Please modify these sections.
- Please add a seprate section for data extration and data item. look at items 9, 10a and 10b of the PRISMA checklist.
- Figure S5, some section of this table are blank? Why?
- Please add reference for each variables. Considering the number of compliments, it is necessary for the readers to know what definition these analyzes have been taken into consideration. In some cases, different definitions can change the results.
- Why you use R software. hese analyzes can be done with RewMan as well.
- Some item are mandatory in reporting a systematic review. For example, flow chart must add to the main text after the first paragraph of the results (study selection, items 16a and b of the PRISMA). Table 1 must include summary characteristics of he included studies and must add to the main text at the end of the second secion of the results (Study characteristics, item 17). The figure 1 is not fix and and its shape is messed up. Please enter as an image.
- Also, you must add the main forst plot to the main text not the supplement.
- Lines 274-287, some statements are related to the methods not the results. Please modified these two section and add some statements to the methods and the remining to the results.
- Discussion section, in the first paragraph of the discussion, a summary of the results of the study should be mentioned, and then the results of the discussion should be discussed in order. We will not compare immediately in the first paragraph.
- Line 307, "the pooled effect estimate of 19 studies indicates ...". This number is incorrect based on the Figure 1 that you merntioned that only 9 article included in the meta-analysis. What is the reason for this inconsistency?
- There is no information related to subgroup analysis in the method section, while it is mentioned in the results and discussion. All results should be based on the work method. In some cases, reporting bias has occurred. Please match all results and methods.
- Lines 381-387, some items you mentioned as strength is one of the requirements of a systematic review and one of the stages of doing it, and it is by no means the strength of your work. Please correct it. You have many weaknesses that are not mentioned. There is a difference between weaknesses and limitations. This section should be corrected. Weaknesses are the things that the researcher could have done, but for whatever reason, he did not do it, but the limitations are the things that should have been done, but it was beyond the responsibility of the researcher. Considering this definition, these items should be modified.
Cheers
Comments on the Quality of English LanguageThere are some type of grammatical errors and punctuation through the text that needs assess by a native editor.
Cheers
Author Response
Dear, Reviewer,
Please see the attachment,
If you face line/lines inconsistency with the main text while reviewing, please find the highlighted text around.

Round 2
Reviewer 1 Report
Comments and Suggestions for Authors
Thank you for your response, which made reviewing the revisions an easy process.
You addressed all of my original suggestions or concerns. I believe the article is nearly ready for publication. I have final thoughts to point out, but I do not think it is necessary for me to review responses or revisions.
1) Under Response 28, you say that you mentioned seasonal effects as a limitation. However, I do not see it as one. I strongly recommend including seasonality in the Discussion at minimum. Including season as an effect modifier would make the paper even stronger, but I do not suggest it is necessary. One middle-ground option could be to include in the Results or Discussion how many of the reviewed articles were conducted in rainy or dry seasons, or both seasons, or did not state.
2) The numbering of figures and tables is still inconsistent throughout the main text and supplementary files. The main text skips numbers. Some tables in the Excel file include titles that do not match their Sheet name. Excel tables sometimes include "S" (e.g., Table S2) and sometimes do not (e.g., Table 1). Figures in the Word document also sometimes include "F" in the number, but not all figures. Please take care in relabelling all figures and tables and ensuring they are correctly referenced in the main text.
Comments on the Quality of English LanguageThe article is written clearly and I believe every sentence is understandable, although some revisions are still required to bring the writing to the standard of a scientific publication.
Reviewer 2 Report
Comments and Suggestions for Authors
Dear Respectable Authors
Thank you for your clarification. Your manuscript is acceptable in this fashion. I wish you the best.
Cheers
Author Response
Thank you very much
I appreciate your in-depth review and suggestions.